# Epidemiology and Diagnostic Perspectives of Dermatophytoses

**DOI:** 10.3390/jof6040310

**Published:** 2020-11-23

**Authors:** Monise Fazolin Petrucelli, Mariana Heinzen de Abreu, Bruna Aline Michelotto Cantelli, Gabriela Gonzalez Segura, Felipe Garcia Nishimura, Tamires Aparecida Bitencourt, Mozart Marins, Ana Lúcia Fachin

**Affiliations:** 1Biotechnology Unit, Unaerp, Av. Costábile Romano, 2201, Ribeirão Preto SP 14096-900, Brazil; mofazolin@gmail.com (M.F.P.); mariheinzen@gmail.com (M.H.d.A.); brucantelli@hotmail.com (B.A.M.C.); gabrielagonzalezsegura@hotmail.com (G.G.S.); felipegnishi@hotmail.com (F.G.N.); tabitencourt@yahoo.com.br (T.A.B.); mmarins@gmb.bio.br (M.M.); 2Department of Genetics, Ribeirão Preto Medical School, University of São Paulo, Ribeirão Preto SP 14049-900, Brazil; 3Department of Biochemistry and Immunology, Ribeirão Preto Medical School, University of São Paulo, Ribeirão Preto SP 14049-900, Brazil

**Keywords:** deep infections, MALDI-TOF MS, mycological diagnosis, PCR, *Trichophyton rubrum*

## Abstract

Dermatophytoses affect about 25% of the world population, and the filamentous fungus *Trichophyton rubrum* is the main causative agent of this group of diseases. Dermatomycoses are caused by pathogenic fungi that generally trigger superficial infections and that feed on keratinized substrates such as skin, hair, and nails. However, there are an increasing number of reports describing dermatophytes that invade deep layers such as the dermis and hypodermis and that can cause deep infections in diabetic and immunocompromised patients, as well as in individuals with immunodeficiency. Despite the high incidence and importance of dermatophytes in clinical mycology, the diagnosis of this type of infection is not always accurate. The conventional methods most commonly used for mycological diagnosis are based on the identification of microbiological and biochemical features. However, in view of the limitations of these conventional methods, molecular diagnostic techniques are increasingly being used because of their higher sensitivity, specificity and rapidity and have become more accessible. The most widely used molecular techniques are conventional PCR, quantitative PCR, multiplex PCR, nested, PCR, PCR-RFLP, and PCR-ELISA. Another promising technique for the identification of microorganisms is the analysis of protein profiles by MALDI-TOF MS. Molecular techniques are promising but it is necessary to improve the quality and availability of the information in genomic and proteomic databases in order to streamline the use of bioinformatics in the identification of dermatophytes of clinical interest.

## 1. Epidemiology of Infections Caused by Dermatophytes

Dermatophytoses are mycoses caused by pathogenic fungi that generally trigger superficial infections in animals and particularly in humans. These dermatophytoses are mainly caused by filamentous fungi that can invade and feed on keratinized substrates [1] such as skin, hair, and nails [2].

The group of dermatophytes comprises 52 keratin-degrading species divided into nine genera: *Trichophyton*, *Microsporum*, *Epidermophyton*, *Arthroderma*, *Lopophyton, Nannizia*, *Ctenomyces*, *Guarromyces*, and *Paraphyton* [3]. Among the fungi causing dermatophytoses in humans, the filamentous fungus *Trichophyton rubrum* is the main causative agent of cutaneous infections of the feet, nails, and body [4].

Infections caused by dermatophytes are called “tineas” and are classified according to the affected site, as shown in Table 1. Infections of the scalp, called tinea capitis, are more common in children, while other tineas more commonly affect postpubertal individuals. The hormonal changes that occur after puberty induce the secretion of acids by sebaceous glands, which, in turn, reduce the incidence of tinea capitis but not of other mycoses. Onychomycoses prevalence increases with, for instance, age. [5]

Dermatophytoses affect individuals worldwide but their incidence is higher in tropical countries because of high temperatures and humidity [12]. Factors that influence the development of dermatophytoses include age, sex, season of year, socioeconomic and cultural conditions, and geographic location [13]. It is estimated that about 10 to 15% of individuals are contaminated with dermatophytes at some point in their life [14]. According to data from the World Health Organization (WHO), dermatophytoses affect about 25% of the world population [15] and 30 to 70% of adults are asymptomatic carriers of these diseases. In Western countries, 80 to 90% of onychomycosis cases are primarily caused by dermatophytes, with 5–17% being due to yeasts and 2–3% to non-dermatophyte molds. In southern European countries, dermatophytes are the causative agents of 40 to 68% of cases, with 21–55% being due to yeasts. In Asian and Middle Eastern countries, dermatophytes account for 40 to 48% of cases, with 43–46% of infections being caused by yeasts and 8–11% by non-dermatophyte molds. Comparatively, in Africa, onychomycosis-related infections are predominantly caused by yeasts [16]. The high prevalence of nail infections in North America is largely due to immigration of dermatophytes from other regions of the world such as West Africa and Southeast Asia. The global prevalence of tinea pedis is estimated at 5.5%, accounting for 50% of all cases of nail disease [17].

Among fungal infections that affect the nails, almost 50% are caused by dermatophytes [18]. It is estimated that tinea pedis is present in 30–70% of the world population. The condition is caused mainly by *T. rubrum*, *T. mentagrophytes* var. *interdigitale* and *Epidermophyton floccosum*, and men are more affected than women [19]. The main causative agent of tinea corporis is *T. rubrum*, which is responsible for 80 to 90% of infections. This prevalence is even higher in men and in children younger than 15 years [20,21]. The risk of onychomycosis increases with age, probably due to the presence of diabetes, poor peripheral circulation, longer exposure to pathogenic fungi, repeated nail trauma, and suboptimal immune function. A family history of onychomycosis is another risk factor [16].

*Trichophyton rubrum* and *E. floccosum* are dermatophytes found worldwide, with *T. rubrum* being the most common species [22]. In addition, other dermatophytes are found in specific regions such as *Microsporum ferrugineum* in Asia, *T. megninii* in Europe, and *T. concentricum* in South America. Despite the specificity of these species in each region, *T*. *rubrum* continues to be the main causative agent, except for Africa where a variety of dermatophytes exist in each microregion, with *Microsporum audouinii* and *T. soudanense* being the most common species [4,23,24].

It is common to find in the literature several articles with epidemiological data from tropical regions. In Brazil, epidemiological data show a higher incidence of infections caused by *T. rubrum* in the southern and southeastern regions, followed by *M. canis* and *T. mentagrophytes*. On the other hand, a higher prevalence of *T. tonsurans*, *T. rubrum,* and *M. canis* is observed in the northeastern region [25,26,27].

Another example are countries on the African continent, where particularly Ethiopia, has a high number of cases of dermatophytosis, not only due to the climatic factor, but also because it is a country with low socioeconomic status, low level of hygiene and health care, and overcrowding [28,29,30]). India is also a country from which several articles are published, and which goes through the same problems as Ethiopia [31,32,33].

## 2. Deep Infection Caused by Dermatophytes

Dermatophytes generally cause superficial infections; however, there are an increasing number of reports describing dermatophytes that invade deep layers such as the dermis and hypodermis and that can spread to the lymph nodes, brain, and bloodstream, causing deep infections [34]. Some conditions are necessary for the establishment of this infection in the patient, including an immunosuppressive state, obesity, diabetes, and advanced age, in addition to other health problems [35]. Other factors related to this disease are environmental factors and a genetic predisposition [36].

Table 2 describes cases of deep infections caused by dermatophytes, highlighting the importance of the correct diagnosis and treatment of these infections.

## 3. CARD 9 Mutation and Fungal Infections

One protein that has attracted attention in invasive infections is CARD9 because of the correlation between mutations in this protein and severe infections caused by dermatophytes [45]. CARD9 is a signaling protein with a caspase recruitment domain that plays a key role in the innate and adaptive immunity. The CARD9 adapter has been shown to enable pattern recognition receptors to induce NF-κB and MAPK activation, an event that triggers a cascade of inflammatory cytokines and provides effective protection against microbial invasion, especially fungal infection [46]. Furthermore, [47] described the case of an individual with skin lesions and localized dermatitis on the neck and face. A skin biopsy revealed the presence of septate and pigmented hyphae characteristic of Phaeohyphomycosis. DNA was extracted from the patient because of the absence of a history of comorbidities and CARD9 deficiency was detected.

## 4. Diagnostic and Identification Methods for Dermatophytoses: from Conventional to Molecular Methods

Diagnosis relies on the culture (positive or negative) and microscopy (distinct direct microscopy and histology, positive or negative as well). Cultures can then be identified using microscopy, MALDI-TOF MS or DNA sequencing. Molecular assays based on DNA (PCR) can provide both diagnosis and identification directly from clinical samples, depending on the assay.

### 4.1. Culture, Microscopy, Histology and Molecular Assays Based on DNA

The conventional methods most commonly used in mycology laboratories for the diagnosis of keratinophilic fungi are based on colony characteristics, microscopic morphology [48], growth requirements, and the results of physiological and biochemical tests. Morphological parameters are the colony pattern, pigmentation, and growth rate. Physiological tests evaluate the hydrolysis of urea and in vitro hair perforation capacity of the fungus. Biochemical parameters include the assimilation of sorbitol [49]. The analysis of physiological and biochemical parameters is useful to distinguish *T. rubrum* from *T. mentagrophytes*, as described by [49].

The conventional identification and diagnosis of dermatophytosis involves techniques of culture and morphological identification of the fungus or the detection of fungal elements by direct microscopy of clinical specimens. Sabouraud dextrose agar or potato dextrose agar is used for the isolation and identification of dermatophytes based in macroscopic characteristics such as colony color and surface. The medium is supplemented with antibiotics and cycloheximide, a reagent specific for the selection of this group of fungi. [17,48,50].

Direct microscopy can be used as a low-cost, rapid result, and an easy-to-do office diagnostic test. However, this test lacks sensitivity and can be enhanced with the combination with histology testing, using periodic acid-Schiff (PAS), for example [17].

The isolation followed by identification by direct microscopic or histology of dermatophytes was considered as a gold standard method. However, this method is limited depending on the quality of the collected sample. For correct isolation of the infectious agent, asepsis of the lesion with 70% alcohol is necessary to eliminate other contaminants, as well as adequate techniques for collection of the affected tissue [48,51]. Consequently, the false-negative rate of this diagnostic method is high, about 35%. In addition, it does not permit to determine the causative fungal species and its viability [17]. In the case of dermatophytes, morphological similarity and the limited or absent sporulation of some species hinder identification by these conventional methods. In addition to these factors, the culture of these species for diagnosis is a time-consuming process, with the growth of some fungi taking up to 6 weeks, delaying the identification of the causative species and the initiation of appropriate therapy [48].

In dermatophytoses, the isolation and correct identification of the causative agent are essential for the choice of adequate treatment. For example, infections caused by anthropophilic dermatophytes require shorter treatment than those caused by zoophilic dermatophyte species. On the other hand, non-dermatophyte fungal species may not respond adequately to the treatment used for dermatophytoses [48].

Some non-dermatophyte species such as *Fusarium*, *Acremonium,* and *Aspergillus* were identified as causative agents of onychomycoses in patients who did not respond to treatment with terbinafine or itraconazole [52]. These antifungal agents have been well established for the treatment of onychomycoses caused by dermatophytes. The wrong diagnosis or lack of identification of the causative agent in these patients may have led to the incorrect hypothesis of the existence of resistant dermatophyte strains in these cases [48]. Thus, the correct identification of the causative fungal species will help to more accurately determine the appropriate treatment, reducing cases of recurrent infections, in addition to contributing to the investigation and control of epidemics [17,48].

Considering the limitations of conventional methods for the identification and diagnosis of dermatophytes, molecular diagnostic techniques are increasingly being used because of their higher sensitivity and specificity and the shorter time necessary for identification of the causative agent. In addition, these methods have become more accessible. Molecular assays based on DNA can provide both diagnosis and identification of the infectious agent directly from clinical samples. Rapid and accurate identification of dermatophytes is essential for the successful treatment of patients with these infections [48].

Molecular diagnostic methods involve the extraction of DNA from the fungus present in clinical samples. The genetic material can be extracted by the traditional phenol-chloroform method or with commercial DNA extraction kits, which provide more efficient extraction. The presence of keratin in clinical sample requires digestion of this protein by enzymatic or non-enzymatic methods [48,53]. DNA extraction followed by the application of different methods for molecular diagnosis has advanced significantly in the last 15 years [48].

DNA sequences are very useful for this purpose and permit an accurate identification. The internal transcribed spacer polymorphisms ITS1 and ITS2 that flank the region encoding the 5.8S rDNA show adequate and reliable sensitivity in distinguishing different species. In addition, the 28S rDNA sequences and genes encoding topoisomerase II and chitin synthase I are used for the identification of dermatophyte species [53].

Molecular diagnostic methods that provide good sensitivity and specificity as well as fast results include the polymerase chain reaction (PCR) and real-time PCR techniques. Studies using PCR reported a rate of correct diagnoses of 74–100% [54].

Conventional PCR is the most widely used method as it provides the best cost–benefit ratio, as well as rapid detection. This technique uses primers amplifying DNA sequences that are specific for the species to be identified. Specific primers to detect any dermatophyte species in a sample (pan-dermatophyte primers) or primers that detect any fungal species (pan-fungal primers) can also be used [53]. The results are interpreted by analyzing the size of the amplicons generated on agarose gel [48,53]. In the study of [55] comparing the performance of different methods for the diagnosis of onychomycosis in 60 patients, the best sensitivity (> 90%) was observed for PCR. In a similar study, [56] suggested the use of PCR as a complementary method for clinical confirmation of suspected cases of onychomycosis. In addition, [57] observed that PCR reduced the time necessary for the diagnosis of dermatophytosis from 4 weeks to 7 h when conventional culture growth was compared with the molecular method, confirming the rapidity and efficacy of PCR.

A post-test called nested PCR can also be used to increase the sensitivity in dermatophyte detection during the diagnosis by conventional PCR [48,57]. This test is a modification of the conventional PCR technique and consists of a second amplification of the fragment amplified by conventional PCR. In this method, the fragment amplified in the first PCR is used as a template for the second amplification, employing primers that only flank the region of the fragment amplified in the first reaction. The aim of this post-PCR technique is to reduce the nonspecific binding of products generated in the first PCR, increasing specificity and sensitivity [48]. Using pan-dermatophyte primers, [58] performed nested PCR for the amplification of translation elongation factor 1-α (Tef-1α) in order to increase the accuracy of identification of relevant dermatophytes in samples of animals with dermatophytosis. Although they increase the sensitivity of conventional PCR, the disadvantage associated with post-PCR techniques is the high risk of contamination of the already amplified PCR products due to the manipulation of these fragments in nested PCR, in addition to prolonging the time to diagnosis [48,53].

Another variant of the PCR technique called multiplex PCR allows the identification of multiple targets in a single reaction, with the benefit of saving time and speeding up the diagnosis. However, for the success of multiplex PCR, it is important to evaluate the possibility of dimerization between the primers used in order to prevent nonspecific amplification. This variant of conventional PCR increases the efficiency of detection and reduces the risk of false-negative results. Highlighted the efficiency, sensitivity, and specificity of multiplex PCR for a faster diagnosis of onychomycosis caused by dermatophytes and by *Fusarium* spp [59]. Similarly, recommended the use of multiplex PCR for the diagnosis of dermatophytic onychomycosis in cases with a negative culture or when the culture is contaminated with fast-growing fungi, a fact that renders the identification of the causal agent problematic [60].

The combination of the restriction fragment length polymorphism (RFLP) technique and PCR can also be used for the identification and discrimination of fungal species in dermatological samples. The so-called PCR-RFLP technique consists of the amplification of specific nucleic acid fragments that exhibit small genetic variations, followed by restriction enzyme analysis and identification of the fragments by gel electrophoresis [48]. This technique can also be used to investigate the composition of microbial communities in different ecological systems such as water and soil [53].

Studying 35 patients with a suspicion of onychomycosis [61] obtained 85.71% sensitivity using PCR-RFLP as the diagnostic test, suggesting that this method is more accurate in the diagnosis of onychomycosis than conventional culture. Used PCR-RFLP to evaluate inter- and intraspecific genomic variations among clinically important dermatophytes isolated from clinical samples [62]. 

The advantages of PCR-RFLP include its low cost and easy design, as well as the fact that no sophisticated equipment is required. However, it is a laborious technique that involves the use of restriction enzymes and identification and knowledge of the desired genetic variants, requiring more time for analysis. Thus, PCR-RFLP is generally not used for routine diagnostics [48].

An alternative assay with a higher diagnostic sensitivity than analysis by gel electrophoresis [53] is PCR-enzyme-linked immunosorbent assay (ELISA). This is a hybrid technique of PCR and ELISA; however, instead of detecting proteins as in a traditional ELISA, PCR-ELISA permits the direct detection of nucleic acids amplified by PCR [48]. This method was developed to increase the sensitivity of detection of dermatophytes in infected tissues and nails. For diagnosis, a PCR amplification with digoxygenin is first performed, generating PCR products labeled with digoxigenin that bind to biotin-labeled oligonucleotides. The biotinylated PCR products are immobilized on a microplate and are detected with anti-digoxigenin. Alternatively, hybridization of the PCR products to fluorescein-labeled oligonucleotide probes can also be performed, followed by detection with horseradish peroxidase-conjugated anti-fluorescein antibodies [48,53].

PCR-ELISA with amplification of the topoisomerase II gene and detection by hybridization using digoxigenin-labeled probes resulted in the successful identification of dermatophyte species such as *T. rubrum*, *T. interdigitale*, *T. violaceum*, *M. canis*, and *E. floccosum* directly in clinical samples within 24 h [53,63]. Dermatophytes of the genus *Trichophyton* were also successfully identified by PCR-ELISA in epidemiological studies conducted in India using skin scrapings from 201 patients [64]. A commercial kit for the detection of dermatophytes in cases of onychomycosis, called Onychodiag, was developed based on the PCR-ELISA technique. However, this technique is not used as a routine laboratory test because of the need for elaborate manipulations and the longer time to diagnosis [48,53].

A very promising method to minimize the risks of contamination and to eliminate the need for post-PCR tests and gel electrophoresis analysis of the results is real-time PCR. This method is also used for the diagnosis of dermatophytosis [48,53]. Real-time PCR detects the presence of DNA or RNA, thus enabling the rapid identification especially of microorganisms. In addition to detecting infectious agents, this technique also provides quantitative data regarding the number of microorganisms present in the sample. Analysis of the results is based on the threshold cycle value (Ct), which corresponds to the number of PCR amplification cycles at which the fluorescence generated within a reaction crosses the fluorescence threshold. Thus, the higher the amount of genetic material present in the sample, the smaller the number of cycles necessary for a positive result and the greater the emitted fluorescence; consequently, low Ct values will be obtained [65].

The diagnostic results obtained by real-time PCR are usually reported as positive, negative, or indeterminate [65]. The quantitative nature of the method in estimating the fungal load of a sample, for example, helps with the differentiation between infection and contamination of clinical samples based on the threshold values obtained in cases of infection. Within this context, studies suggest that real-time PCR can be applied successfully for the diagnosis of dermatophytosis with much higher precision compared to classical diagnostic methods [48]. Furthermore, evaluating dermatophytes in nail samples, [66] showed that real-time PCR detected the presence of dermatophytes in samples that tested negative by conventional PCR, suggesting a higher diagnostic sensitivity of the former.

In an attempt to increase the diagnostic response, real-time PCR can be performed as described by [67], who indicates multiplex RT-PCR as the fastest and most efficient method for the identification of dermatophyte species in clinical samples, or using pan-dermatophyte primers as described by [66] for the diagnosis of dermatophytes in nail samples. Commercial kits designed to facilitate the identification of dermatophytes by real-time PCR have also been developed. One example is the Derma Genius^®^ multiplex kit for the accurate and rapid identification of *T. rubrum*, *T. interdigitale,* and *Candida albicans* in nails [68].

Taken together, it is evident that the development of molecular techniques has brought several benefits, permitting a better diagnosis of dermatophytoses and more appropriate treatment for patients with these diseases.

### 4.2. MALDI-TOF-MS

Another promising technique for the identification of microorganisms is the fingerprint of a protein extract by matrix-assisted laser desorption/ionization time of flight mass spectrometry (MALDI-TOF MS). This method is applied to the identification of a wide range of species [69,70,71,72,73,74], including dermatophytes [69,70,71,72,73,74,75,76,77,78] but until now, is not used in clinical diagnostic method for dermatophytosis. In addition, this approach has drastically reduced the response times in routine clinical laboratories [72,74,77,79]. MALDI-TOF MS uses the characteristic fingerprint of a protein to identify a specific microorganism by combining species–specific protein patterns included in a comprehensive reference spectra library [80]. Hence, the identity of microorganisms can be established at the species level in mycology [75,81].

In the case of fungi in general, identification by MALDI-TOF MS can be compromised if the sample is contaminated with culture medium, especially when fungal colonies are present and cannot be separated from the agar [79]. This limitation is also discussed by [82], which suggested the use of a new medium called Id-Fungi plates (IDFP) from Conidia^®^.

The use of this technique for the study of dermatophytes is more complex because of the variable phenotypes of these fungi, which can result in variations of the protein spectra; this heterogeneity is affected by the growth conditions and by the mycelial zone examined (concomitant presence of different fungal structures, hyphae and/or conidia, in the same culture) [83,84]. However, the main limitation for the identification of dermatophytes is the inadequate representation of dermatophyte species in the reference spectra libraries of the current commercial MALDI-TOF MS identification systems. Identification by MALDI-TOF MS still requires the successful growth of fungus in culture. However, even in the case of adequate sampling, dermatophytes exhibit relative low sensitivity, with approximately 30% of the culture results being false negative [85,86].

The main concern associated with the identification of pathogenic fungi by MALDI-TOF MS are the similarities of molecular components, which can render sister species indistinguishable, for example, *Trichophyton rubrum* and *T. violaceum or Microsporum canis* and *M. audouinii* [70,71,78,87]. For each MALDI-TOF MS system, the reference database for species coverage is essential for the approach, as long as the database used in the analysis of MALDI-TOF MS is improved to include adequate spectra of different strains of each species studied [88]. Within this context, laboratories are recommended to generate and complement the mass spectra for their main local species or lineages and to register them with commercial reference libraries [89].

In contrast to bacteria, fungi require additional processing steps to break the cell wall, extract proteins, and inactivate the isolate [90]. These additional processing steps and limited libraries have led laboratories to develop their own methodologies and databases in an attempt to overcome the barriers to the adoption of the technique [70,74,91]. The time, media and culture conditions used for the strain to be identified must be the same as those used as a reference in the database used for comparison and spectral identification so that there are no changes in the microorganism protein profiles [70,92]. These limitations have prevented the widespread implementation of this technology in clinical laboratories for the identification of filamentous fungi.

Another limitation of MALDI-TOF MS is that, unlike publicly available sequence databases such as GenBank, commercial MALDI-TOF MS databases are usually exclusive to companies. Although the low identification rates for some organisms may be increased by user addition of mass spectral entries of under-represented species or strains (to cover intraspecies variability), or even by the re-addition of reference strain spectra to the library, especially those created using parallel growth conditions and preparation methods, doing so may be beyond the capacity of some laboratories [93]. A non-commercial option is available through an online identification tool that comprises more than 900 species of fungi and more than 200 genera of fungi, which allows the identification of fungi, including dermatophytes, for free thus facilitating the identification of fungi [94].

Four different commercial MALDI-TOF MS platforms for the routine identification of fungi in clinical microbiology laboratories are currently available. These platforms include the Andromas (Andromas SAS, Paris, France), Axima@ Saramis (Shimadzu/AnagnosTec, Duisburg, Germany), Bruker Biotyper (Bruker Daltonics, Bremen, Germany), and Vitek MS (bioMérieux, Marcy l’Etoile, France) systems [95].

The reported identification rates of dermatophytes by MALDI-TOF MS range from 13.5% to 100% [79]. However, the efficiency of this technique depends primarily on the standard library provided by the manufacturer or a supplementary library of the manufacturer. The use of these standard libraries supports the reported variable findings, for example, expansion of the library improves the accuracy of identification of dermatophytes by MALDI-TOF MS [70,78,79]. Effective identification of dermatophyte species depends on the number and variety of isolates available in the reference spectral library [70,72,78,79].

Table 3 summarizes the main advantages and disadvantages of each method for the identification and diagnosis of the dermatophytes discussed in this article.

## 5. The Potential of Molecular Approaches for the Study of Dermatophytes

A vast range of molecular approaches exist in clinical mycology. These approaches are not only limited to diagnostic tests but can also be used for prospecting new antifungal compounds for the treatment of these diseases, as done by [97,98,99], as well as for the identification of conserved regions that contribute to the virulence of these pathogens as described by [100], enabling the discovery of new therapeutic targets. Moreover, technologies for large-scale transcriptome analyses such as RNAseq and microarray have contributed to a better understanding of dermatophyte-host interactions at the molecular level [101,102], in addition to providing new insight into the effect of commercial antifungals such as terbinafine on species such as *T. rubrum*. Modulation of 277 differentially expressed genes was observed in co-cultures of the dermatophyte *T. rubrum* with the HaCat keratinocyte cell line in the presence of terbinafine. Approximately 28% of these genes have been studied in other dermatophytes but not in *T. rubrum*, thus contributing to the elucidation of the functions of these genes in this species and of the response mechanisms to antifungal agents. In addition, modulation of different genes with an important role in the biosynthesis and transport of ergosterol, such as ERG1, ERG5, ERG11, CYP51, and CYP61, was demonstrated [103].

## 6. Future Perspectives

Major advances have been made over the years in the identification and diagnosis of dermatophytes. However, some limitations, such as the lack of a complete database for the identification of pathogenic fungi, are still an obstacle. The establishment of an extensive online database containing the molecular profiles of fungal species involved in human diseases, with the possibility of inclusion of additional information by users, would be a major advance in this field. This requires the development of reliable algorithms that allow the rapid consultation of thousands of reference molecular profiles and easy access for everyone.

## Figures and Tables

**Table 1 jof-06-00310-t001:** Main types of “tineas”.

Tinea	Main Dermatophyte	Site of Infection	Reference
**Tinea corporis**	*Trichophyton rubrum* *T. tonsurans* *Microsporum canis*	Body (chest, face, arms, and/or legs)	[6]
**Tinea pedis** **(athlete’s foot)**	*T rubrum* *T. interdigitale* *Epidermophyton floccosum*	Foot (soles or interdigital spaces)	[7]
**Tinea capitis**	*T. tonsurans* *Microsporum canis* *Trichophyton violaceum* *Trichophyton soudanense*	Scalp	[8,9]
**Tinea cruris**	*T. rubrum*	Groin folds	[10]
**Tinea unguium** **(onychomycosis)**	*T. rubrum* *T. interdigitale*	Nails	[11]

**Table 2 jof-06-00310-t002:** Case reports of deep infections caused by dermatophytes.

Comorbidity	Symptoms	Reference
Hepatitis C and liver cirrhosis	Subcutaneous nodules	[37]
Human immunodeficiency virus (HIV)	Atypical, multiple or extensive lesions	[38]
Transplant and chemotherapy patients	Nodules	[39]
History of diabetes mellitus	Palpable nodules in the ankle	[40]
Patients without immunodeficiency, history of onychomycosis	Erythema, papules and nodules in the submandibular area, neck, and chest	[41]
Use of immunosuppressive drugs	Fungal abscesses	[42]
Patients without immunodeficiency, hypertension and angina	Red spots with secretion	[43]
Fungal infection for 2 years, onychomycosis	Loss of vision	[44]

**Table 3 jof-06-00310-t003:** Main advantages and disadvantages of the identification and diagnostic methods used for dermatophytes.

Methods of Identification/Diagnosis	Advantages	Disadvantages	Reference
Conventional (culture followed by direct microscopy or histology)	Low cost of materials Well-established standard method	Time-consuming process to obtain the result High false-negative rate	[48,51,96]
Conventional PCR	Better cost–benefit ratio Rapid detection	Post-PCR tests might be necessary to complement the diagnosis	[53]
Nested PCR	Reduces nonspecific binding of PCR products More specific results Good sensitivity	High risk of contamination Prolongs the time to diagnosis	[48,53]
Multiplex PCR	Identification of multiple targets in the same reaction Time saving and rapid diagnosis Higher detection efficacy Reduces the risk of false-negative results	Possibility of nonspecific binding between primers	[48]
PCR-RFLP	Low cost Does not require sophisticated equipment	Use of restriction enzymes is necessary Requires more time for diagnostic analysis Not commonly used for diagnosis	[48]
PCR-ELISA	Higher sensitivity than techniques that use analysis by gel electrophoresis	Elaborate manipulations are necessary Requires more time for diagnostic analysis Not commonly used for diagnosis	[48,53]
Real-time PCR	Low risk of contamination Post-PCR tests are not required Rapid identification Quantitative detection	Specific equipment is necessary Higher cost than conventional PCR	[48,53,65]
MALDI- TOF MS	Identification of the microorganism at the genus, species, and strain level	Difficulty of access or incomplete information of some dermatophyte species in databases used for identification Sister species might be indistinguishable because of similar molecular components	[70,71,72,74,75,77,79,81,85,86,87,93]

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
