# Peer review of "Epidemiology and Diagnostic Perspectives of Dermatophytoses"

_jof, 2020, doi:10.3390/jof6040310_

Round 1

Reviewer 1 Report

The review by Petrucelli et al. combines the epidemiology and diagnosis/identification of dermatophytoses and their causative agents. This quality review is well presented and provides adequate information on this topic. Some minor remarks should however be addressed before publication.

  • To my opinion, there is some confusion between diagnosis and identification. Diagnosis relies on the culture (positive or negative) and microscopy (of which I would distinct direct microscopy and histology, positive or negative as well). Cultures can then be identified using microscopy, MALDI-TOF MS or DNA sequencing. Molecular assays based on DNA that are described (e.g. PCR) can provide both diagnosis and identification directly from clinical samples, depending on the assay. This should be clarified.
  • The same comment can be done for Table 3 which should be adapted accordingly. In particular, conventional diagnosis include positive/negative culture and positive/negative microscopy (direct examination and histology). But identification of the causative species is well possible based on microscopy. MALDI-TOF is an identification method, not a diagnosis method.
  • Line 43: nails, not “hails”
  • Lines 46-48: hormonal changes at puberty reduces the incidence of tinea capitis but not of other mycoses. Onychomycoses prevalence increases with age for instance.
  • Table 1: T. violaceum and T. soudanense are other important agents of tinea capitis. Moreover, T. interdigitale belongs to the T. mentagrophytes complex. Within this complex, T. interdigitale is responsible for most cases of tinea pedis. But mentioning T. mentagrophytes (sensu stricto) in addition to T. interdigitale is confusing/inappropriate. I would delete T. mentagrophytes for tinea pedis causative agents. For tinea unguium: please correct the spelling of T. interdigitale.
  • Lines 65-66: the first sentence of the paragraph refers to nails infection while the second sentence is about tinea pedis… Tinea pedis and onychomycosis are different infections.
  • Lines 79-94: it is a bit strange to detail the case of Brazil. I understand that the authors are from Brazil but from an international point of view, there is no rationale to highlight the case of Brazil more than another country.
  • Line 95: there is an increasing number
  • Lines 110-113: if that patient suffered from CARD9 deficiency, he was not healthy but well immunocompromised! Besides, the fungal agent was not identified in this publication and the infection could be caused by a non-dermatophyte species.
  • Line 114: as mentioned, the word “diagnosis” should be used correctly. Please adapt the title
  • Lines 128-129: again, culture (positive/negative) is a diagnostic method but identification of the culture based on microscopy is possible.
  • Line 167: primers amplifying DNA sequences
  • Lines 204-205: I would delete the last sentence of the paragraph. It is out of the scope of this review
  • Line 266: fingerprint of a protein extract.
  • Line 268 + Table 3: MALDI-TOF MS does not allow typing of fungal strains. Identification is limited to the species level in mycology.
  • Line 271: about the difficult separation of the colonies from the agar: please mention the publication by Sacheli et al (http://dx.doi.org/10.1111/myc.13156) to address this problem.
  • Line 288: it is unclear what “register” means in this sentence. Do you mean “compile”?
  • Lines 289-294: this paragraph should be rephrased. Opposite to bacteria and yeasts, filamentous fungi indeed require a full extraction for MALDI-TOF. This increases the workload but it is not really a limitation. Protocols attempting to decrease this workload were not particularly successful. What is important to mention is that culture media and conditions used for the strain to be identified should be the same than the media and culture conditions used to build the reference spectra of the database used for spectral comparison and identification.
  • Line 312: a non-commercial MALDI-TOF database for fungi (including dermatophytes) is available through an online identification tool. This database is actually the most extended with more than 900 species represented. Please mention this database. See notably the following article: https://doi.org/10.1128/JCM.00263-17

Author Response

The review by Petrucelli et al. combines the epidemiology and diagnosis/identification of dermatophytoses and their causative agents. This quality review is well presented and provides adequate information on this topic. Some minor remarks should however be addressed before publication.

  • To my opinion, there is some confusion between diagnosis and identification. Diagnosis relies on the culture (positive or negative) and microscopy (of which I would distinct direct microscopy and histology, positive or negative as well). Cultures can then be identified using microscopy, MALDI-TOF MS or DNA sequencing. Molecular assays based on DNA that are described (e.g. PCR) can provide both diagnosis and identification directly from clinical samples, depending on the assay. This should be clarified.

The authors agree with reviewer’s suggestion. Please see the modifications in the new version of manuscript (please see lines 119-122).

  • The same comment can be done for Table 3 which should be adapted accordingly. In particular, conventional diagnosis include positive/negative culture and positive/negative microscopy (direct examination and histology). But identification of the causative species is well possible based on microscopy. MALDI-TOF is an identification method, not a diagnosis method.

The authors agree with reviewer’s suggestion. Please see the modifications in the new version of manuscript (Table 3).

  • Line 43: nails, not “hails. ”ok please line 44
  • Lines 46-48: hormonal changes at puberty reduces the incidence of tinea capitis but not of other mycoses. Onychomycoses prevalence increases with age for instance. Please see line 48-50
  • Table 1: T. violaceum and T. soudanense are other important agents of tinea capitis. Moreover, T. interdigitale belongs to the T. mentagrophytes complex. Within this complex, T. interdigitale is responsible for most cases of tinea pedis. But mentioning T. mentagrophytes (sensu stricto) in addition to T. interdigitale is confusing/inappropriate. I would delete T. mentagrophytes for tinea pedis causative agents. For tinea unguium: please correct the spelling of T. interdigitale.

The authors agree with reviewer’s suggestion. Please see the modifications in the new version of manuscript (table 1 line 51).

  • Lines 65-66: the first sentence of the paragraph refers to nails infection while the second sentence is about tinea pedis… Tinea pedis and onychomycosis are different infections. The authors agree with reviewer’s suggestion. Please see lines 65-68
  • Lines 79-94: it is a bit strange to detail the case of Brazil. I understand that the authors are from Brazil but from an international point of view, there is no rationale to highlight the case of Brazil more than another country. The authors agree with reviewer’s suggestion. The Brazil epidemiology part was Please see lines 85-88
  • Line 95: there is an increasing number. The authors agree with reviewer’s suggestion. Please see line 96
  • Lines 110-113: if that patient suffered from CARD9 deficiency, he was not healthy but well immunocompromised! Besides, the fungal agent was not identified in this publication and the infection could be caused by a non-dermatophyte species.

The authors agree with reviewer’s suggestion. Please see line 113-116.

  • Line 114: as mentioned, the word “diagnosis” should be used correctly. Please adapt the title
  • Lines 128-129: again, culture (positive/negative) is a diagnostic method but identification of the culture based on microscopy is possible.
  • Line 167: primers amplifying DNA sequences
  • Lines 204-205: I would delete the last sentence of the paragraph. It is out of the scope of this review
  • Line 266: fingerprint of a protein extract.
  • Line 268 + Table 3: MALDI-TOF MS does not allow typing of fungal strains. Identification is limited to the species level in mycology.

Line 271: about the difficult separation of the colonies from the agar: please mention the publication by Sacheli et al (http://dx.doi.org/10.1111/myc.13156) to address this problem.

The authors agree with reviewer’s suggestion and this reference was included in the manuscript.

  • Line 288: it is unclear what “register” means in this sentence. Do you mean “compile”?
  • Lines 289-294: this paragraph should be rephrased. Opposite to bacteria and yeasts, filamentous fungi indeed require a full extraction for MALDI-TOF. This increases the workload but it is not really a limitation. Protocols attempting to decrease this workload were not particularly successful. What is important to mention is that culture media and conditions used for the strain to be identified should be the same than the media and culture conditions used to build the reference spectra of the database used for spectral comparison and identification.

The authors agree with reviewer’s suggestion. Please see the modifications in the new version of manuscript

  • Line 312: a non-commercial MALDI-TOF database for fungi (including dermatophytes) is available through an online identification tool. This database is actually the most extended with more than 900 species represented. Please mention this database. See notably the following article: https://doi.org/10.1128/JCM.00263-17

The authors agree with reviewer’s suggestion. Please see the modifications in the new version of manuscript (lines- 117-347).

Reviewer 2 Report

Interesting and well-written article. The authors do not describe the individual methods, but present their advantages and disadvantages. There is no such review in the literature, and the writing style is innovative. 

Just a few suggestions:

  • I propose to separate more paragraphs so that the reader can find it easier in the text. It is especially needed when describing methods, there is clearly no indication in which part of what types of methods are presented.
  • Epidemiology is too focused on Brazil. Probably it is natural due to the origin of the authors, however, it is necessary to complete the data from other places in the world for the information to be reliable.
  • In the abstract, keranized must be changed to keratinized
  • Information on lines 48 contradicts those on lines 85 and 99. Older age reduces or increases the risk of dermatophytosis? This has to be clearly assessed

Author Response

Comments and Suggestions for Authors

Interesting and well-written article. The authors do not describe the individual methods, but present their advantages and disadvantages. There is no such review in the literature, and the writing style is innovative. 

Just a few suggestions:

  • I propose to separate more paragraphs so that the reader can find it easier in the text. It is especially needed when describing methods, there is clearly no indication in which part of what types of methods are presented.
  • . Please see the modifications in the new version of manuscript
  • Epidemiology is too focused on Brazil. Probably it is natural due to the origin of the authors, however, it is necessary to complete the data from other places in the world for the information to be reliable.
  • Please see the modifications in the new version of manuscript. Please see lines 84-88. Articles of others countries were added  Please see lines 89-93
  • In the abstract, keranized must be changed to keratinized.
  • Please see line 17